# Assessment of ADHD Subtypes Using Motion Tracking Recognition Based on Stroop Color–Word Tests

**DOI:** 10.3390/s24020323

**Published:** 2024-01-05

**Authors:** Chao Li, David Delgado-Gómez, Aaron Sujar, Ping Wang, Marina Martin-Moratinos, Marcos Bella-Fernández, Antonio Eduardo Masó-Besga, Inmaculada Peñuelas-Calvo, Juan Ardoy-Cuadros, Paula Hernández-Liebo, Hilario Blasco-Fontecilla

**Affiliations:** 1Faculty of Medicine, Autonomous University of Madrid, 28029 Madrid, Spain; 2Department of Psychiatry, Puerta de Hierro University Hospital, 28222 Majadahonda, Spain; 3Department of Statistics, University Carlos III of Madrid, 28903 Getafe, Spain; 4School of Computer Engineering, University Rey Juan Carlos, 28933 Madrid, Spain; 5Department of Psychology, Comillas Pontifical University, 28015 Madrid, Spain; 6Department of Psychology, Autonomous University of Madrid, 28029 Madrid, Spain; 7Airbus Group Inc., 28901 Getafe, Spain; 8Department of Child and Adolescent Psychiatry, University Hospital 12 de Octubre, 28041 Madrid, Spain; 9Health Sciences College, Rey Juan Carlos University, 28933 Madrid, Spain; 10Department of Psychiatry, Marqués de Valdecilla University Hospital, University of Cantabria, 39008 Santander, Spain; 11Center of Biomedical Network Research on Mental Health (CIBERSAM), 28029 Madrid, Spain; 12UNIR-ITEI & Health Sciences School, International University of La Rioja, 26006 Logroño, Spain

**Keywords:** ADHD, subgroups, Stroop effect, motion tracking, reaction time, Kinect

## Abstract

Attention-Deficit/Hyperactivity Disorder (ADHD) is a neurodevelopmental disorder known for its significant heterogeneity and varied symptom presentation. Describing the different subtypes as predominantly inattentive (ADHD–I), combined (ADHD–C), and hyperactive–impulsive (ADHD–H) relies primarily on clinical observations, which can be subjective. To address the need for more objective diagnostic methods, this pilot study implemented a Microsoft Kinect-based Stroop Color–Word Test (KSWCT) with the objective of investigating the potential differences in executive function and motor control between different subtypes in a group of children and adolescents with ADHD. A series of linear mixture modeling were used to encompass the performance accuracy, reaction times, and extraneous movements during the tests. Our findings suggested that age plays a critical role, and older subjects showed improvements in KSWCT performance; however, no significant divergence in activity level between the subtypes (ADHD–I and ADHD–H/C) was established. Patients with ADHD–H/C showed tendencies toward deficits in motor planning and executive control, exhibited by shorter reaction times for incorrect responses and more difficulty suppressing erroneous responses. This study provides preliminary evidence of unique executive characteristics among ADHD subtypes, advances our understanding of the heterogeneity of the disorder, and lays the foundation for the development of refined and objective diagnostic tools for ADHD.

## 1. Introduction

Attention-Deficit/Hyperactivity Disorder (ADHD) is a common neurodevelopmental disorder in childhood, affecting about 7% of the population [1]. Its main symptoms are inattention, hyperactivity, and impulsivity. People with ADHD may have executive dysfunction, attention deficit, and performance impairments in cognitive domains such as working memory, planning, cognitive flexibility, inhibition of motor response, and interference control. These impairments can negatively affect academic, occupational, and social functioning [2,3]. ADHD is associated with abnormalities in the dorsal frontal striatum, orbitofrontal striatum, and prefrontal cortex of the pontine-cerebellar circuit, affecting cognitive control, reward processing, and time perception [4]. According to the Diagnostic and Statistical Manual of Mental Disorders, Fifth Edition (DSM-5), there are three subtypes of ADHD that are predominantly inattentive (ADHD–I), hyperactive–impulsive (ADHD–H), or combined presentation (ADHD–C) [5]. Boys are twice as likely as girls to develop ADHD [6]. ADHD–I is the most common subtype, with a prevalence rate ranging from 38–57% of all ADHD patients, while ADHD–H and ADHD–C account for 19–37% [7].

The diagnosis of ADHD still relies heavily on the observation of physicians, families, and teachers, which can be influenced by various factors, potentially impacting the final diagnosis [8,9]. The main diagnostic criteria are based on DSM-5, and the diagnostic categories remain non-homogeneous, with mechanistic heterogeneity and varying neuropsychological subgroups within each subtype [10,11,12]. Even individuals with the same subtype can exhibit significant differences in cognitive deficits and executive dysfunction [11]. ADHD shares etiology and pathophysiology with other psychiatric conditions, affecting individual sensitivity to pharmacotherapy and psychotherapy [13,14]. More recently, a framework of integrated developmental psychopathology (DP) and the National Institute of Mental Health Domain Criteria (RDoC) has attempted to refine theories of ADHD etiology, which is still under development and may explain heterogeneity at the levels of behavioral and neural circuits [15]. However, the conventional approach still depends on DSM-5-based self-reported questionnaires and symptom reports and lacks objective tests such as neuroimaging or biomarkers [8].

Although individuals with ADHD and typical controls exhibit noticeable differences in performance in numerous tests, there is still considerable controversy surrounding the outcomes of these tests [11,16,17,18,19,20,21]. Some studies suggest that inhibition and variation in reaction times are core features of ADHD, although others question this view [16,17,19]. All these differences may be attributed only to specific subsets [18] and that children diagnosed with ADHD might exhibit comparable accuracy to typical controls in basic psychological testing tasks [22]. With the increasing popularity of computerized testing, numerous tests such as TOVA, Conners’ CPT, Flanker, Go/no-Go, and Stroop have been applied to diagnose and explore ADHD [18,19,23]. Computerization has improved the accuracy, consistency, and standardization of the experimental performance of ADHD, and has allowed for reliable measurement of cognitive functions, such as reaction times and processing speed, which can be difficult to accurately measure with pen and paper tests [19,23]. Although many neuropsychological measures have been proposed to be related to ADHD, each measure only applies to a subset of individuals with the disorder [11,18]. Subtypes of ADHD form a significant source of heterogeneity to be considered [13]. ADHD–I exhibits poorer interference inhibition than other subtypes, while ADHD–C and ADHD–H demonstrate poorer short-term memory/working memory and greater variability in reaction time [17,19,20]. According to a study on mothers of patients, children with ADHD–C and ADHD–H are more challenging to raise than children with ADHD–I [24].

Studies have looked at detecting physiological patterns and high-resolution motor characteristics in daily tasks and psychological tests for children with ADHD [25,26]. Children with ADHD have shown inferior motor proficiency in various domains compared to control groups, including gross and fine motor skills, visual-motor control, running speed, agility, balance, and bilateral coordination [27]. High-resolution motor features have captured different movement patterns in children with ADHD [22,28,29,30,31]. A study found that using mouse movements instead of the keypress method during a Go/no-Go task enhanced the mean response times’ difference and provided more variance in groups with high and low ADHD scores [32]. Changing the response model to a more sophisticated approach may provide more detail and facilitate the exploration of performance patterns in patients with ADHD. Some studies find that children with ADHD have higher levels of physical activity, measured by wearable devices equipped with gyroscopes, than control groups during school classes [25,28,30]. Children with ADHD also exhibit greater motor neural activity in various body parts in TOVA tests and have higher multi-joint kinesthetic activity in studies using the Kinect device [29,31]. Furthermore, a study assessed ADHD in children and found a correlation between reaction time and impulsivity via a Kinect-based CPT [33]. The use of high-resolution motor features with conventional executive function tests has the potential to enhance the distinctions between various subtypes of ADHD, thus providing more objective benchmarks for diagnosis.

The main objective of this pilot study is to reveal differences between the different subtypes of ADHD using a body motion perception system based on a Kinect device and the Stroop Color–Word test. Specifically, our goal is to test whether patients with the inattentive subtype present a distinctive executive profile compared to the combined/predominant hyperactive–impulsive subtypes. By examining high-resolution motor characteristics and standard executive characteristics, we hope that the characteristics could provide richer objective information to help clinicians have a more objective understanding of the status and evolution of ADHD patients.

## 2. Materials and Methods

### 2.1. Participants

The current study included 61 children and adolescents aged 8 to 17 years recruited from the Department of Child and Adolescent Mental Health at the Puerta de Hierro University Hospital in Majadahonda, Spain, between 5 January 2020 and 15 September 2021. Of these, 30 were diagnosed with the ADHD–C type, 7 with the ADHD–H type, and 24 with the ADHD–I type. All included patients were diagnosed according to the DSM-5 criteria and diagnosed by clinical psychologists or psychiatric physicians. The collection of sociodemographic and clinical information involved the use of electronic medical records (EMRs) and interviews with legal guardians and patients. All participants had normal or corrected-to-normal vision. Exclusion criteria include neurological disease history, head injuries, epilepsy, or other psychiatric difficulties that could affect the Stroop tests. Patients with physical defects such as limb defects, Congenital Myasthenic Syndrome, Congenital Muscular Dystrophy, or other physical problems that could affect the performance on Stroop tests have also been excluded. Additionally, individuals with a Wechsler intelligence test IQ score below 80 assessed in a previous visit or limb defects were excluded from the study. Patients with learning disorders, but with developmental dyscalculia that had no difficulty reading or speaking, were also included in the sample. All participants and their legal guardians gave their informed consent in writing before participating in the study. The study was approved by the Ethics Committee of the Puerta de Hierro University Hospital in Majadahonda (Madrid, Spain) on 2 December 2019 (PI 178/19).

### 2.2. Basic Measures

In Table 1, the main sociodemographic and clinical characteristics and the comparison between the two groups of ADHD are displayed. Some physical features that may affect the test performance, such as the preference of the dominant hand and the use of glasses, are also reported in this table. Since all patients had IQs above 80, we reported that patients with IQs greater than 130 were intellectually gifted according to concepts from previous studies [34]. Medication therapy was recorded to indicate whether participants were on medication at the time of testing. This refers to the intake of prescribed ADHD medications (stimulants or non-stimulants) on the day of the test for those subjects who were medically managed for their ADHD. The participants took medicine on the day of testing, if they were on medication for their ADHD. Psychological treatment indicated whether the patient had received any type of ADHD psychotherapy currently or in the past three months. The comorbidities and psychological dysfunction of the patients are shown in Table 2.

### 2.3. Kinect Stroop Color–Word Test

The Stroop Color Word Test (SCWT) is one of the most widely used tests to assess cognitive function and inhibitory control, and in the current study, we used a computerized version adapted from the Golden version [35,36]. It consists of three tasks: Word, Color, and Color–Word, each subtest lasting 45 s. The Word test requires participants to name a series of black words. In the Color test, the task is a series of colored blocks or “X” marks and requires participants to name their colors. The Color–Word test involves color words displayed in mismatched colors, demanding color naming while inhibiting reading [35,37]. Participants must complete as many words as possible on each task.

Kinect (Microsoft Corporation, model: Kinect for Windows v2, released on 15 July 2014) is a device developed by Microsoft that could be used to track movement and record voice. The device includes a standard RGB color camera, infrared projector, sensor, and microphone array. Powered by a set of free Kinect software development kits (SDKs, version 2.0.1410), Kinect detects and tracks the motion of 25 major body joints at 34–35 milliseconds per frame [31,33,38]. The Microsoft Kinect-based Stroop Color–Word Test (KSCWT) program used in this study was developed in C-sharp language. Response zones were formatted with three color blocks on the two sides of the shoulders and above the head, corresponding to the colors used in SCWT: green, blue, and red. The patients responded by raising their wrists to the corresponding color block on the shoulders and above the head. The test included three subtests lasting 45 s (see Figure 1A–C).

The experiment was conducted in a quiet, soundproof room with the patient, the legal guardian, and the experimental assistant. The legal guardian was instructed to remain silent during each subtest procedure. The Kinect device was mounted on an adjustable stand and the patients were positioned 2.5–3 m directly in front of the device. Before the test, the legal guardian and experimental assistant located themselves in the operating control area behind the Kinect device. The operator terminal displayed real-time visual images of the patient and mapped joint locations. If necessary, the position of the patient and the Kinect device could be adjusted to optimize joint detection and motion recognition. The test display was projected onto the screen behind the Kinect device.

Before each subtest begins, an introduction with text and graphic descriptions is displayed on the screen, describing the content and requirements of the task, and the experimental assistant explains the content to the patient. Patients were asked to lift their arms from a neutral position to the colored blocks above their head and shoulders, and then return their arms to the neutral position after the stimulus appeared. Patients could view their real-time body contour map on the screen and the location of the response color blocks (see Figure 1A–C). The position of the red block above the head was automatically adjusted to the patient’s height. The positions of the green and blue blocks on both sides of the shoulders were symmetrically adjusted horizontally before each subtest by the experimental assistant to ensure that they were not too far or too close to the shoulders of the patients, making it more comfortable for the patients to respond to each stimulus.

After the experiment started, the stimulus was displayed on the screen and Kinect tracked the joint movements to detect where the wrist reached the color patch. Once a response is detected, subsequent stimuli are presented. Two consecutive stimuli are always different. The computer recorded various parameters, including stimulus occurrence timestamps, corresponding colors and words, correct and incorrect answers, response timestamps, colors and words, and the 3D motion trajectories of 25 joints.

## 3. Data Extraction and Analysis

### 3.1. Data Extraction

The main data extracted included the number of responses, accuracy rates, reaction time (RT), and response hand. We calculate two types of RT: unadjusted RT, which measures the time from the stimulus to the start of the wrist movement, and adjusted RT, which measures the time from the stimulus onset to the Kinect detecting the response. The adjusted RT is closer to the traditional SCWT on a computer [39]. To calculate the adjusted reaction time, a Kalman filter was used to smooth the overall motion trajectory, followed by a gradient descent algorithm [38,40]. In the cases of multiple peaks, the peak closest to the stimulus and an arm lift distance greater than 10 cm was used as the adjusted RT (as shown in “Peak0” in Figure 1E). Patients were allowed to use either hand to respond to the stimulus. For each stimulus item, the participant ID, responding hand, stimulus presentation time, response outcome (correct/incorrect), and adjusted unadjusted RT were estimated.

The hand that did not respond to the stimulus could also have extraneous movement during the tests. From the Kinect data, we extracted the vector angular velocity of the hand that did not respond to the stimuli. In the stimulus-response period of each item, after the withdrawal phase of the hand was removed, the sum of vector angular tracking was considered the Extraneous Movement Score (EMS).

### 3.2. Data Analysis

As an exploratory study, our study focuses on the difference between the types of ADHD–I and ADHD–H/C and the potential factors that affect performance in ADHD children and adolescents. In the univariate analysis, the basic demographic and clinical characteristics of the two groups were compared. General measures extracted from the SCWT and Kinect regarding the performance of the two ADHD groups were compared, as well as different subtests of the SCWT. We also analyzed the ex-Gaussian characteristics (mu, sigma, and lambda) of adjusted and unadjusted RTs. Fisher’s exact test, Student’s *t* test, and the Wilcoxon rank sum test were used in the univariate analysis.

To explore factors that may affect patient accuracy, multivariate linear models or mixed linear models were established for further analysis. Also, to explore differences between individuals and groups, with patients as random effects, a series of mixed linear models were built to fit the unadjusted and adjusted RT of each response item and its characteristics (response hand, stimulus presentation time, response outcome, etc.). We use rigorous statistical techniques to mitigate potential biases, sensitivity, post hoc, and subgroup analyses are also applied if necessary [41,42]. All analyzes were performed using Python (version 3.10) [43] and R (version 4.2) [44]. Python was used for data extraction and plotting, and R was used for univariate and multivariate modeling.

## 4. Results

### 4.1. Clinical and Demographic Characteristics

The study included 61 patients with ADHD, 24 in the ADHD–I group and 37 in the ADHD–H/C group (see Table 1). The patients in the ADHD–I group were approximately 1.5 years older than those in the ADHD–H/C group. The mean age at the time of initial diagnosis was significantly higher in the ADHD–I group compared to the ADHD–H/C group. Seventy-three percent (n = 27) of the patients in the ADHD–H/C group received pharmaceutical treatment, while only 38% (n = 9) of the patients in the ADHD–I group received treatment.

Executive dysfunction was the most common comorbidity, affecting approximately 41% (n = 25) of all patients, with 54% (n = 13) of the patients in the ADHD–I group and 32% in the ADHD–H/C group having this dysfunction. None of the patients in the ADHD–I group had comorbid oppositional defiant disorder or conduct disorder; in the ADHD–H/C group, only 8% (n = 3) of the patients in the ADHD–H/C group had ODD and 14% (n = 5) had conduct disorder (see Table 2).

There were few differences in the overall performance between the two groups in the SCWT. Regardless of whether the data was unadjusted or adjusted, there were no differences in the number of responses or the average reaction time (see Table 3). The results of the Ex-Gaussian distribution also did not provide any clear metrics to differentiate between the two groups. The main differences were the correct rate and the standard deviation of reaction times. The ADHD–I group had a significantly lower standard deviation in mean reaction time than the ADHD–H/C group in the Color–Word test in adjusted RT (U=582, p=0.04 and Cohen’s d=0.54). Comparable results were seen in the other two subtests, but only close to the statistical significance threshold value (see Figure 2). For unadjusted RT, the Color–Word test showed near-significant results (U=567, p=0.07, and Cohen’s d=0.49) (see Figure 2).

### 4.2. Correct Rate

In order to comprehensively evaluate the impact of other factors on the correct rate, a linear mixed effects model was used to analyze the impact of ADHD grouping, patient age, medication status, and Stroop subtest on the correct rate. As shown in Appendix A, the age and Stroop subtests had a significant effect on the correct rate. The estimated marginal mean correct rate in the ADHD–I group is approximately 92.49% (95%CI = 86.84–98.04), and in the ADHD–H/C group, it is 89.70 percent (95%CI = 83.68–95.72), meaning there were no significant differences between the two groups (p=0.37). For each additional year of age, the correct rate increased by approximately 1.18 percent (95%CI = 0.23–2.12, p=0.04). Among the subtests, the Color–Word test had a lower correct rate (M=88.78%, 95%CI = 83.50–94.05) compared to the Word test (M=92.99%, 95%CI = 87.72–98.27) and the Color test (M=91.51%, 95%CI = 86.23–96.89), where the differences were only statistically significant between the Color–Word and Word tests (p=0.04) but not in the Color–Word and Color tests (p=0.27).

Several analyzes were performed to examine the interactions between the two groups and various factors. Patients without medical treatment did not show significant differences in the correct rate of both ADHD groups (W=568.5, p=0.22). However, among patients receiving medication, the correct rate of ADHD–C/H (M=88.56, 95%CI = 84.55–92.58) was lower than that of ADHD–I (M=97.22, 95%CI = 90.27–104.17, W=102, p=0.001). However, these differences are not significant after taking into account the influence of age and subtests. Although the Word test and the Color–Word test differed between the two groups of ADHD (see Figure 3), these differences did not gain the same importance due to the effects of age and medication treatment (see the Marginal Mean Subtest Patterns in Figure 3). After considering the effects of the treatment status and subtests, the age still significantly affected the correct rate (β=1.54±0.66, df=55, t=2.34, p=0.02). The estimated correct rate in the ADHD–I group was 92.87% (95%CI = 87.12–98.62), and the ADHD–H/C group was 88.53% (95%CI = 83.55–79.52), where the difference between the two groups were insignificant (p=0.33; see the Marginal Mean Age Patterns in Figure 3).

### 4.3. Reaction Times

In the realm of measuring the overall performance of the reaction time, as shown in Table 3, only the standard deviation of the mean RT in the Color–Word test observed significant differences between the two groups of ADHD. To gain a deeper understanding of the influence of potential factors that affect RT and their impacts on the two groups of ADHD, we incorporated a comprehensive analysis using multiple linear mixed-effects models. The variables were derived from the attributes of each stimulus item, including associated subtests, stimulus presentation time, response hand, and correctness of the response. Furthermore, demographic variables that showed imbalances between the two groups of ADHD (age and medication treatment) were included. Despite a disparity in diagnostic age between ADHD groups, we considered the correlation between diagnostic age and current age (r=0.70, 95%CI = 0.54–0.81, p<0.001), and only current age was used as an analytical factor. The model’s random effects component duly accounted for individual variations contingent upon stimulus presentation time and subtests.

Focus on unadjusted reaction times across the three Stroop subtests. After considering the individual difference, the estimated mean RT in the Word test is 1258 milliseconds (95%CI= 1169–1346), while the Color test shows a significantly lower RT (M=1129, 95%CI= 1041–1217, p<0.001). The Color–Word test shows a higher RT (M=1303, 95%CI= 1215–1392) than the Color test (p<0.001), but the difference with the Word test is not significant (p=0.13). In adjusted RT, the Color test RT (M=744, 95%CI = 696–793) remains the lowest, and there are differences with both the Word test (M=867, 95%CI= 817–916, p<0.001) and the Color–Word test (M=912, 95%CI = 862–961, p=0.008).

In view of possible variations in individual response time stability during each subtest, we also examined the interaction model between ADHD groups, subtests, and test times (indexed by stimulus presentation time) (Figure 4). In unadjusted RT analyses, the interaction between these three factors does not show a general significance (F=1.29, p=0.28). However, the interaction between the ADHD groups and the Stroop subtests showed some differences (F=3.25, p=0.04). In particular, the ADHD–H/C group showed a higher estimated average RT of 1283 milliseconds (95%CI = 1168–1379) in the Word test compared to the Color test (M=1125, 95%CI = 1012–1238, p<0.001), where similar results were observed in the Color–Word test (M=1346, 95%CI = 1232–1460, p<0.001), but the difference between the Word and Color–Word test was not statistically significant (p=0.73). The estimated difference in the unadjusted RT between the subtests in the ADHD–I group was not statistically significant; also, the difference between the ADHD groups in the same subtest was not statistically significant. In adjusted RT, the interaction between these three factors was significant (F=5.15, p=0.006). For the ADHD–H/C group, the reaction time of the Color–Word test is 943 (95%CI = 880–1006), higher than that of the Color test (M=748, 95%CI = 685–811, p<0.001) and the Word test (M=879, 95%CI = 816–943, p=0.03). In the ADHD–I group, the reaction time in the Color–Word test was 861 milliseconds (95%CI = 782–940), almost identical to that of the Word test (M=853, 95%CI = 774–932, p=0.98), but compared to the Color tests, a significant difference was observed (M=738, 95%CI = 660–816, p<0.001).

After controlling for individual differences, several factors influence the reaction time, both in adjusted and unadjusted RT, including age, stimulus presentation time, response accuracy (correct/incorrect), and subtesting. Age has a robust impact on reaction times (see Figure 5). In unadjusted RT, for every one-year increase in age, in general, the RT decreases by 90 milliseconds (95%CI = 61–119, p<0.001). In the ADHD–I group, this decrease in RT was reduced to 42 milliseconds (95%CI = 13–72, p=0.007), and in the ADHD–H/C group, it is 89 milliseconds (95%CI = 64–114, p<0.001). However, the overall age-based RT difference between the two groups was not significant (p=0.15). According to adjusted RT, reaction times decreased by 31 ms for each additional year of age (95%CI = 15–46, p<0.001). In the ADHD–I group, the decrease for each year old is 20 milliseconds (95%CI = −3–44, p=0.10), and in the ADHD–H/C group, it is 38 milliseconds (95%CI = 17–58, p<0.001). Similarly, age-based RT differences between the two groups of ADHD were also not significant (p=0.37).

Medication treatment significantly increased the reaction time in unadjusted RT. For the entire sample, the treatment with the medication increased the reaction times by approximately 128 milliseconds (95%CI = 1–255, p=0.05). There was no significant difference between patients with ADHD–I who received or did not receive medication (t=2.83, p=0.18), but significant differences were observed between patients with ADHD–H/C with or without medication therapy (t=3.19, p=0.004). However, there were no significant differences in RT between the two ADHD groups under the same treatment conditions, regardless of where they were without medication therapy (t=1.62, p=0.37) or those who had medication therapy (t=0.67, p=0.91). In the case of adjusted RT, medication therapy did not affect RT (t=0.97, p=0.33); also, there were no differences between specific groups of ADHD (see Figure 5).

For correct responses, the unadjusted RT was estimated to increase on average by 368 milliseconds (95%CI = 283–453, p<0.001). Specifically, the ADHD–H/C group had an estimated mean RT of 1193 milliseconds (95%CI = 1091–1295) for correct responses, while incorrect responses took only 800 milliseconds (95%CI = 664–935, see Figure 5). Among patients in the ADHD–I group, a correct answer required an RT of 1292 milliseconds (95%CI = 1179–1406), while an incorrect answer required only 1009 milliseconds (95%CI = 802–1215). The difference between correct and incorrect responses in the same ADHD–H/C group is significant (t=7.96, p<0.001), and similar results were found in the same ADHD–I group (t=3.13, p=0.01). However, neither in the correct responses nor in the incorrect responses, the difference between the two ADHD groups (ADHD–I versus ADHD–H/C) was significant (p=0.55 for the correct responses, p=0.34 for the incorrect responses). In the adjusted reaction times, the mean estimated increase in reaction time for correct responses was 242 milliseconds (95%CI = 187–296, p<0.001). The difference between correct and incorrect responses was significant in the ADHD–H/C group (t=8.91, p<0.001), but not in the ADHD–I group (t=2.06, p=0.17). The RT of incorrect responses in the ADHD–I group was significantly higher than in the ADHD–H/C group (t=2.76, p=0.03). The unadjusted RT increased by 86 milliseconds when using the right hand to answer (95%CI = 48–123, p<0.001), and this difference is also significant in the ADHD–H/C group (t=3.68, p=0.008). In other conditions and the adjusted RT, the effect of the response hand was not statistically significant.

### 4.4. Extraneous Movement Score

The Extraneous Movement Score (EMS) is used to estimate the hyperactivity level during the test. The mean EMS are 30.79 (SD=53.10), but the EMS distribution showed positive skewness (3.99) and kurtosis (23.42). Patients with ADHD–H/C have a mean EMS of 32.48 (SD=57.72), and the ADHD–I group had 28.21 (SD=45.04). The influence of age on EMS was obvious: in patients under 10 years of age, they had significantly higher EMS scores (p<0.001) and SD EMS (p<0.001) than those aged 10 and older (see Appendix A). In contrast, the mean and SD of EMS in the two ADHD groups are not significantly different (U=493, p=0.5 and U=486, p=0.5, respectively).

We also developed linear mixed-effects models that accounted for individual differences to further explore factors that influence EMS. We found that certain factors were significant, including subtests (F=7.50, p<0.001), age (F=22.68, p<0.001), stimulus color (F=26.78, p<0.001) and correctness of responses (F=9.79, p=0.002), medication status (F=32.99, p<0.001), test time (F=17.10, p<0.001), and response hand (F=5.10, p=0.02). However, the ADHD groups (F=0.77, p=0.38) do not influence the model. Age remained a robust factor affecting EMS, and combined with other factors such as the Word and Color tests, stimuli with green and blue answers all reduced the extraneous movement scores. The medication therapy, responding with the right hand, and the advancing of the test time increased the movement scores (see Appendix A). No interaction was detected between these factors and the ADHD groups.

## 5. Discussion

This study used a Kinect-based Stroop Color–Word test with motion tracking to explore the performance of subtypes of ADHD, particularly inattentive (ADHD–I), hyperactive–impulsive, or combined presentation (ADHD–H/C). We also compared two patterns of reaction times (unadjusted and adjusted RT) and a movement score (EMS) to estimate the hyperactivity level of patients during the test. The results demonstrate that this Stroop test can provide high-resolution data on the movement of ADHD patients, but it is still difficult to apply to find the difference between different subtypes of ADHD. The impact of SCWT interference on patients with ADHD is complex.

The Stroop interference effect exists in the Stroop test of our study. Age emerges as an essential factor in the performance of ADHD patients. As age increases, the correct rate of the tests increases significantly and reaction times decrease. After taking into account the influence of age and treatment status, there were no significant differences in reaction times and accuracy between the ADHD–I and ADHD–H/C patients. Further analysis revealed that in case of incorrect responses or using the left hand to respond to the stimulus, all patients had shorter RTs, but in incorrect responses, the ADHD–H/C patients had even shorter RTs than ADHD–I patients in adjusted RTs. Patients of major age (over 10 years) had fewer unnecessary hand movements during the Stroop test. However, there were no significant differences in the extraneous movement score between different types of ADHD.

Some studies suggest that age is not related to Stroop interference effects, and patients with ADHD–I are believed to experience more pronounced Stroop interference effects [17,45,46]. In our study, we observed that age does indeed have an impact on Stroop performance. Regardless of whether it is ADHD–I or ADHD–H/C, there is an overall higher accuracy rate, a shorter reaction time, and a smaller hyperactivity score as age increases. However, in terms of overall performance, there were no significant differences between ADHD subgroups based on age changes. This finding is similar to the conclusions of other studies related to RT in ADHD [22,47,48].

The unadjusted RT encompasses the total time of the motor planning and execution phases, while the adjusted RT reflects more of the motor planning aspect. Patients with ADHD–H/C may experience more pronounced challenges with motor planning and executive control, making it more difficult to inhibit erroneous responses. In situations that had incorrect reactions to stimuli, the ADHD–H/C group showed a shorter adjusted RT than ADHD–I. ADHD–H/C patients also showed more pronounced interference effects than ADHD–I patients.

Medication treatment affects reaction times in unadjusted RT, leading to increased RTs. Furthermore, patients in the ADHD–H/C group who were receiving pharmacological treatment on unadjusted RT were significantly longer than patients who did not receive treatment in our study, which was not significant in the adjusted RT or ADHD–I groups. This also confirms the conclusion of the previous report that the ADHD–H/C group has poor motor planning and executive control problems, and medication treatment may improve these conditions, increasing unadjusted RT. The ADHD patients demonstrated a weaker ability to modulate reaction times based on conditions and less motor planning [22]. It is possible that medication treatment has a more pronounced effect on ADHD–H/C, but more information is needed to validate this hypothesis. Currently, limited research discusses the effects of medication treatment in different subtypes of ADHD [49].

We tried to measure hyperactivity in ADHD patients through EMS. A study had shown that ADHD symptoms were significantly higher in children than in adolescents [50]. In our study, patients with major-age ADHD also had a lower EMS, but we did not find any significant differences in the ADHD subtypes. Several studies have concluded that ADHD patients exhibit increased limb activity in daily life, but there is no specific research that examines these aspects based on different subtypes [25,29,30]. Since the current study requires hand movements to respond, EMS could be affected by body movements and the location of the sensing area. More research with an improved design is needed to demonstrate whether high-resolution motor features can differentiate ADHD subtypes via direct observation.

The present study also had several limitations that should be acknowledged. First, the design of the experiment had some aspects that needed improvement, including the display form, response form, and response patterns. Second, this is a primary experiment, the COVID-19 pandemic affected sample collection, and the small sample size and the imbalance between ADHD subgroups in terms of age and treatment status resulted in minimal differences between groups. The rate of intellectual giftedness among our study participants was higher than commonly observed in the general clinical population, which could suggest a selection bias or the unique characteristics of our study cohort. Therefore, future research should extend this study to a broader ADHD population to assess the extent to which our findings can be generalized. Also, the study design introduced additional noise during data processing due to the stimulus intervals and response patterns. In future studies, we should add a break for resetting between stimuli. Future research could benefit from incorporating assessment scales alongside the experimental results, allowing for a more comprehensive evaluation from multiple dimensions. These limitations underscore the need for further investigation and improvement in future studies.

## 6. Conclusions

The study used Stroop color–word testing based on Kinect motion tracking to explore two groups of ADHD (ADHD–I and ADHD–H/C). Innovatively, two reaction time patterns and extraneous movement scores were compared. Age emerges as an essential factor in the performance of ADHD patients. The performance in the SWCT improved as age increased. ADHD patients with hyperactive–impulsive or combined type may have deficits in motor planning and executive control, and these cause them to have shorter reaction times in wrong responses and have more difficulty suppressing erroneous responses, but more studies are still necessary to verify this conclusion. In conclusion, Kinect-based SCWT can offer high-resolution motion tracking to explore the movement characteristics during Stroop tests, which combines the advantages of both SCWT and microbehavior detection. However, an improved experimental design is still needed to identify the different patterns that may exist in different ADHD subtypes.

## Figures and Tables

**Figure 1 sensors-24-00323-f001:**
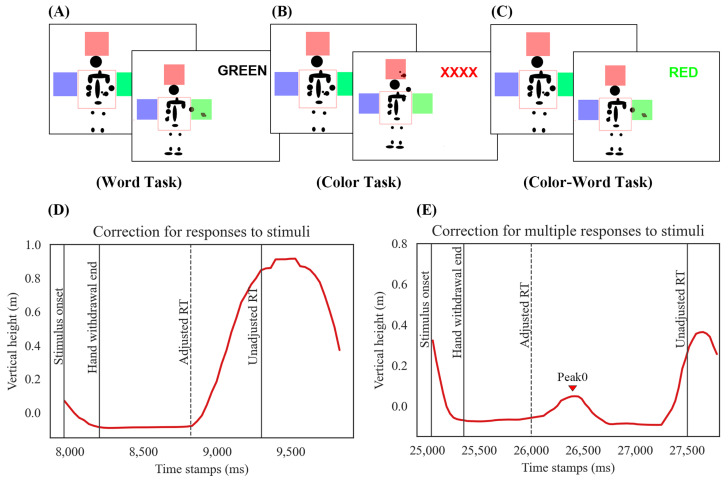
Impact of Reaction Time Variability and Correct Rate in the Two Groups by Subtest and Adjustment Algorithm.

**Figure 2 sensors-24-00323-f002:**
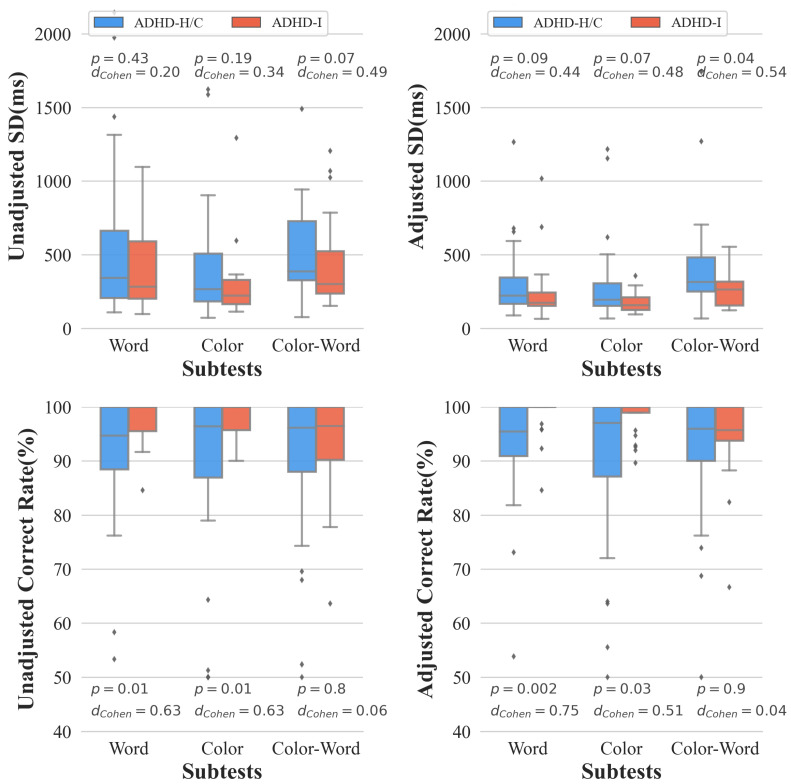
Comparison of Correct Rates and Standard Deviation of Reaction Time in the Gesture-Based SWCT Subtests Using Unadjusted and Adjusted Measures.

**Figure 3 sensors-24-00323-f003:**
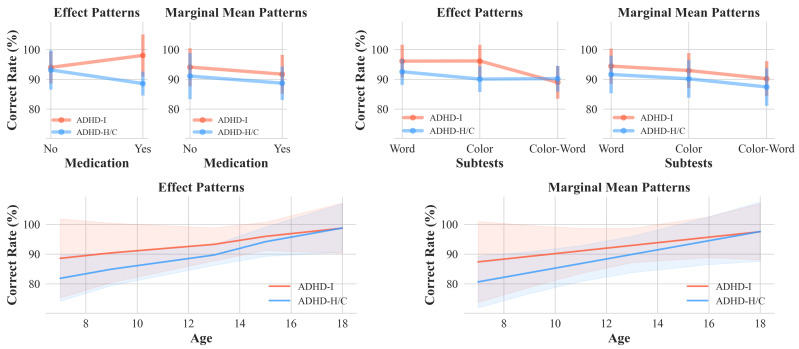
The figure shows the Changes and Confidence Intervals of the Two ADHD Groups by Age, Medication, and Subtests: Use Effect Patterns and Estimated Marginal Mean Patterns.

**Figure 4 sensors-24-00323-f004:**
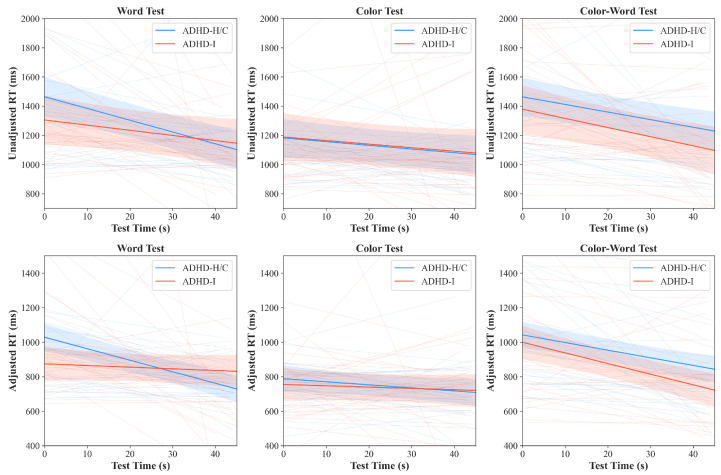
Estimated changes and confidence intervals in unadjusted and adjusted reaction times of the two ADHD groups over test time in different subtests. The thin and semi-transparent lines indicate the changes in each patient. There was considerable variation in reaction times between patients as the test progressed.

**Figure 5 sensors-24-00323-f005:**
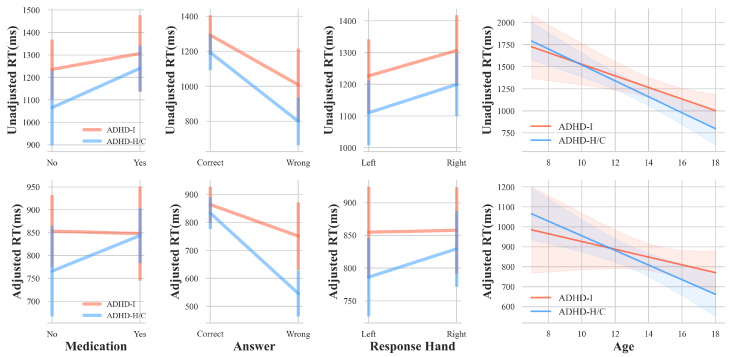
The figure shows estimated Marginal Means and Confidence Intervals of Unadjusted and Adjusted Reaction Times in Two Groups of ADHD by Age, Medication, Correctness, and Response Hand.

**Table 1 sensors-24-00323-t001:** Comparisons between ADHD–I and ADHD–H/C Groups in Demographic Characteristics.

Characteristic ^1^	Overall (n = 61)	ADHD–H/C (n = 37)	ADHD–I (n = 24)	*p*-Value ^2^
Gender, n (%)				>0.9
Male	35 (57.4%)	21 (56.8%)	14 (58.3%)	
Female	26 (42.6%)	16 (43.2%)	10 (41.7%)	
Age	13.41 ± 2.62	12.84 ± 2.65	14.29 ± 2.37	**0.03**
Ethnic, n (%)				0.5
Caucasian	53 (86.9%)	31 (83.8%)	22 (91.7%)	
Others	8 (13.1%)	6 (16.2%)	2 (8.3%)	
Birthplace, n (%)				0.7
Spain	53 (86.9%)	33 (89.2%)	20 (83.3%)	
Others	8 (13.1%)	4 (10.8%)	4 (16.7%)	
Wearing glasses, n (%)	17 (27.9%)	9 (24.3%)	8 (33.3%)	0.6
Left handedness, n (%)	5 (8.2%)	2 (5.4%)	3 (12.5%)	0.4
Intellectual giftedness, n (%)	9 (14.8%)	5 (13.5%)	4 (16.7%)	0.7
Diagnosis age	11.3 ± 3.2	10.7 ± 3.2	12.4 ± 3.1	**0.04**
Medication therapy, n (%)	36 (59.0%)	27 (73.0%)	9 (37.5%)	**0.01**
Treatment duration ^3^	5 (0–24)	12 (0–36)	0 (0–24)	**0.02**
Psychotherapy ^4^, n (%)	33 (54.1%)	21 (56.8%)	12 (50.0%)	0.8

^1^ n (%); Mean (SD); ^2^. Fisher’s exact test and Welch Two Sample *t*-test. ^3^ The duration of treatment is reported in months; only medications related to ADHD were recorded; data are reported as median (IQR); and Wilcoxon
rank sum test was used to estimate the *p*-value. ^4^ Psychotherapy shows the number of patients who received psychotherapy in the last three months.

**Table 2 sensors-24-00323-t002:** Comparison of Comorbid Disorders in the Two Subtypes of ADHD.

Comorbid Disorders ^1^	ADHD–I		ADHD–H/C		TOTAL	
	Sample Num.	Disorder Num. (%)	Sample Num.	Disorder Num. (%)	Sample Num.	Disorder Num. (%)
ASD	24	3 (12.5%)	37	5 (13.5%)	61	8 (13.1%)
Learning Disorders ^2^	24	6 (25.0%)	37	9 (24.3%)	61	15 (24.6%)
Crossed Laterality	24	1 (4.2%)	37	1 (2.7%)	61	2 (3.3%)
S (P)CD	24	1 (4.2%)	37	5 (13.5%)	61	6 (9.8%)
Executive Dysfunction	24	13 (54.2%)	37	12 (32.4%)	61	25 (41.0%)
Emotional Dysregulation	24	1 (4.2%)	37	4 (10.8%)	61	5 (8.2%)
Tics Disorder	24	4 (16.7%)	37	3 (8.1%)	61	7 (11.5%)
Elimination disorder	24	1 (4.2%)	37	1 (2.7%)	61	2 (3.3%)
ODD	24	/	37	3 (8.1%)	61	3 (4.9%)
OCD	24	2 (8.3%)	37	2 (5.4%)	61	4 (6.6%)
Conduct disorder	24	/	37	5 (13.5%)	61	5 (8.2%)
MDD	24	2 (8.3%)	37	2 (5.4%)	61	4 (6.6%)

^1^ ASD: Autism Spectrum Disorder; S(P)CD: Social (Pragmatic) Communication Disorder; OCD: Obsessive Compulsive Disorder; ODD: Oppositional Defiant Disorder; MDD: Major Depressive Disorder. Clinical assessment of comorbidities (i.e., learning disabilities, autism spectrum disorder) and psychological dysfunction (i.e., executive dysfunction, intersectional laterality) was performed based on all data provided by electronic medical records and legal guardians of the patients. Comorbidities were diagnosed according to DSM-5 criteria. ^2^ Only children and adolescents with developmental dyscalculia, but no difficulties in reading and speaking, were included.

**Table 3 sensors-24-00323-t003:** Comparing Kinect-Based Stroop Test Performance in ADHD–I and ADHD–H/C Groups.

Characteristic ^1^	Overall (n = 61)	ADHD–H/C (n = 37)	ADHD–I (n = 24)	*p*-Value ^2^
**Based on Unadjusted RT**				
**Number of responses (n)**				
Word	24 (21–27)	24 (21–27)	24 (20–27)	>0.9
Color	28 (25–31)	28 (25–31)	28 (26–30)	0.3
Color–Word	25 (22–29)	26 (22–29)	25 (22–29)	>0.9
**Correct rate (%)**				
Word	100 (92–100)	95 (88–100)	100 (96–100)	0.01
Color	97 (92–100)	96 (87–100)	100 (96–100)	0.01
Color–Word	96 (88–100)	96 (88–100)	96 (90–100)	0.8
**RT Mean (ms)**				
Word	1158 (1033–1306)	1168 (1062–1306)	1153 (1017–1307)	0.9
Color	1026 (931–1112)	1032 (921–1157)	1015 (942–1101)	0.7
Color–Word	1189 (1031–1423)	1195 (1083–1423)	1141 (1031–1357)	0.5
**RT SD (ms)**				
Word	295 (204–600)	342 (206–662)	282 (201–591)	0.4
Color	263 (179–411)	267 (184–507)	222 (163–328)	0.2
Color–Word	372 (287–627)	387 (327–727)	302 (236–524)	0.07
**RT Ex-Gaussian μ (ms)**				
Word	913 (798–1031)	886 (794–1031)	964 (839–1034)	0.2
Color	815 (713–917)	814 (677–909)	833 (762–925)	0.2
Color–Word	889 (730–1031)	881 (699–1031)	891 (773–1030)	0.8
**RT Ex-Gaussian σ (ms)**				
Word	100 (39–226)	102 (39–232)	96 (46–185)	0.9
Color	78 (54–146)	86 (57–174)	70 (53–139)	0.5
Color–Word	93 (44–267)	101 (32–267)	90 (55–265)	>0.9
**RT Ex-Gaussian λ **				
Word	2 (1–6)	3 (1–7)	2 (1–4)	0.6
Color	2 (1–5)	2 (1–7)	2 (2–3)	0.7
Color–Word	3 (1–7)	3 (1–12)	2 (2–4)	0.6
**Based on Adjusted RT**				
**Number of responses (n)**				
Word	23 (20–26)	23 (20–26)	23 (19–26)	0.7
Color	27 (23–30)	27 (23–30)	26 (23–28)	0.3
Color–Word	23 (20–28)	24 (21–28)	23 (19–26)	0.4
**Correct rate (%)**				
Word	100 (93–100)	95 (91–100)	100 (100–100)	0.002
Color	97 (93–100)	97 (87–100)	100 (99–100)	0.03
Color–Word	96 (91–100)	96 (90–100)	96 (94–100)	0.9
**RT Mean (ms)**				
Word	833 (738–927)	855 (739–915)	823 (731–964)	>0.9
Color	733 (656–821)	697 (652–825)	737 (662–794)	>0.9
Color–Word	847 (740–1009)	859 (740–1022)	801 (742–1006)	0.5
**RT SD (ms)**				
Word	194 (159–291)	222 (166–346)	173 (152–244)	0.09
Color	181 (134–255)	195 (153–305)	158 (126–210)	0.07
Color–Word	289 (202–438)	315 (249–482)	263 (156–317)	0.04
**RT Ex-Gaussian μ (ms)**				
Word	711 (606–797)	682 (571–797)	727 (667–796)	0.3
Color	613 (516–697)	584 (480–694)	652 (552–733)	0.12
Color–Word	656 (565–770)	626 (513–770)	679 (599–762)	0.3
**RT Ex-Gaussian σ (ms)**				
Word	126 (77–202)	132 (70–221)	110 (99–169)	0.6
Color	118 (66–178)	124 (59–189)	117 (69–161)	>0.9
Color–Word	106 (57–250)	132 (82–250)	95 (43–233)	0.3
**RT Ex-Gaussian λ **				
Word	1 (0–2)	1 (0–2)	1 (0–2)	0.9
Color	1 (0.0–1.8)	1 (0.2–2.3)	1 (0.0–1.5)	0.3
Color–Word	2 (1–4)	2 (1–4)	2 (0–5)	0.8

^1^ Number and median (IQR) were used as descriptive statistics. ^2^ Mann–Whitney U test and Fisher’s exact test were used.

## Data Availability

The data that support the findings of this study are available from the corresponding author upon reasonable request. Due to privacy and ethical considerations, some restrictions may apply to the availability of certain data.

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
