# Peer review of "Assessment of ADHD Subtypes Using Motion Tracking Recognition Based on Stroop Color–Word Tests"

_sensors, 2024, doi:10.3390/s24020323_

Round 1

Reviewer 1 Report

Comments and Suggestions for Authors

Thank you for giving me an opportunity to read your research.

The present study, as authors explain, aims to reveal differences between the different subtypes of ADHD using a body motion perception system and to test whether patients with the inattentive subtype present a distinctive executive profile compared to the combined / predominant hiperactive-impulsive subtypes.

The article addresses a relevant topic and research study employs adequate method in collecting the data, strict adherence to statistical good practice and detailed and comprehensive presentation of the results. The authors discuss the result concluding that although collecting high-resolution data on the movement of participants it was difficult to find the difference between subtypes of ADHD. Analysis of factors in evaluating the performance of ADHD patients age showed significance, with correct rate of test increased with age. Limitations of the study are described.

My only recommendation for improvement is for the abstract, the more precise objective of research should be stated (as in the text further on), adding information on analyses performed and more clear indication of findings regarding both difference between subtypes and factors effecting performance. This is important for readers to know what to expect precisely.

Author Response

Firstly, we would like to thank the Editor and the Reviewer for their valuable time. We explain below how the reviewer's comments have been included in the revised version of the manuscript.

We thank the reviewer for this supportive comment.

Thank you for your valuable suggestion. We have carefully considered your recommendations and have revised the abstract accordingly. The new abstract shows the objectives, the type of analysis performed, and the findings obtained.

Reviewer 2 Report

Comments and Suggestions for Authors

This is an excellent paper. In the field of ADHD, there have been many attempts to find a test which would help to identify ADHD. There are some available, but each has its limitations. In this study, the authors research a Kinect Microsoft motion detection device to help in the evaluation of children with ADHD; and they test if this test can distinguish between the inattentive subtype and the hyperactive/impulsive/combined subtype. Although their study doesn't find that this test can do that; they have several interesting findings. Furthermore, they contribute to an important area of research which hopefully moves the field forward toward finding a test or model of testing that will help with research and clinical care of ADHD patients. 

I found the introduction/background thorough and helpful; I found the methods section thorough and detailed. The results section was detailed with good figures; and the conclusions and limitations discussed were excellent. 

My only suggestions are quite small: 

1) in lines 107-110, the authors define what is meant by medication in this testing. Although it becomes more clear later in the paper, readers may benefit from more clear language that the participants took medicine on the day of testing, if they were on medication for their ADHD.

2) In Table 1, the rate of intellectual giftedness was high overall and in both groups - with the overall percentage at 14.8%. I suggest this should be discussed in the limitations, as a general clinical population would not have almost 15% of the individuals in the intellectually gifted range. 

3) On lines 105-106, the authors write that all patients had IQs above 80, and some had intellectual giftedness. On lines 421-422, the authors discuss that one of the limitations is that intelligence tests could be helpful, as well as ADHD symptom scales. The comment about intelligence tests is confusing, as they document that each patient had an IQ test already, or we wouldn't know that they all had IQs above 80, with about 15% meeting criteria for giftedness. 

Apart from these small suggestions, I support accepting this paper. 

Author Response

Firstly, we would like to thank the Editor and the Reviewer for their valuable time. We explain below how the reviewer's comments have been included in the revised version of the manuscript. We thank the reviewer for this supportive comment.

Point-by-point response to Comments

Comments 1: in lines 107-110, the authors define what is meant by medication in this testing. Although it becomes more clear later in the paper, readers may benefit from more clear language that the participants took medicine on the day of testing, if they were on medication for their ADHD.

Response 1: We have included this clarification in lines 122-126.

“Medication therapy was recorded to indicate whether participants were on medication at the time of testing. This refers to the intake of prescribed ADHD medications (stimulants or non-stimulants) on the day of the test for those subjects who were medically managed for their ADHD. The participants took medicine on the day of testing, if they were on medication for their ADHD.”

Comments 2: In Table 1, the rate of intellectual giftedness was high overall and in both groups - with the overall percentage at 14.8%. I suggest this should be discussed in the limitations, as a general clinical population would not have almost 15% of the individuals in the intellectually gifted range.

Response 1: According to the reviewer's comment, the following paragraph has been included in lines 435-439.

“The rate of intellectual giftedness among our study participants was higher than commonly observed in the general clinical population, which could suggest a selection bias or the unique characteristics of our study cohort. Therefore, future research should extend this study to a broader ADHD population to assess the extent to which our findings can be generalized."

Comments 3: On lines 105-106, the authors write that all patients had IQs above 80, and some had intellectual giftedness. On lines 421-422, the authors discuss that one of the limitations is that intelligence tests could be helpful, as well as ADHD symptom scales. The comment about intelligence tests is confusing, as they document that each patient had an IQ test already, or we wouldn't know that they all had IQs above 80, with about 15% meeting criteria for giftedness.

Response 3:

All variables mentioned in the article were collected on the day of the Kinect test except the intelligence test which was administered at a previous visit. We have clarified this (line 110) and deleted previous lines 421-422 which, as the reviewer indicates, leads to confusion.

Once again, we thank the reviewer for his/her time to improve our work and for his/her supportive words.

Reviewer 3 Report

Comments and Suggestions for Authors

This article is well-writen and easy to read. However, there are still some things I would suggest that authors address:

1) P.2: The main diagnostic criteria are based on DSM-5, and the diagnostic 32 categories remain non-homogeneous, with mechanistic heterogeneity and varying neuropsychological subgroups within each subtype. - While there is no doubt that DSM-V is commonly used in diagnosing ADHD, recently much attention has been focused on Research Domain Criteria (RDoC). I think the authors should at least mention RDOC approach (see Musser & Raiker, 2019 for review)

2). P2: Although individuals with ADHD and typical controls exhibit noticeable differences in performance in numerous tests, there is still considerable controversy surrounding the outcomes of these tests. - It is not clear which controversy the authors are referring to here.

3) P.2:  By examining high-resolution activity characteristics and standard executive characteristics - I am not sure I understand the distinction between "activity" and "executive" characteristic. In my opinion, substituting "motor" for "activity" could enhance the clarity of the paper.

4) P.3: Please clarify which groups are you comparing in Table 1 (perhaps in the table title).

4) P.8: Please clarify which groups are you comparing in Table 3 (perhaps in the table title). 

5) Figure 7: What do the error bars represent? Please clarify, preferably in the figure caption.

6) Generally speaking, the authors' argument can be improved by incorporating other research that deals with hand motion and ADHD, such as Leontyev et al., 2018.

Leontyev, A., Sun, S., Wolfe, M., & Yamauchi, T. (2018). Augmented Go/No-Go task: Mouse cursor motion measures improve ADHD symptom assessment in healthy college students. Frontiers in psychology9, 496.

Author Response

Firstly, we would like to thank the Editor and the Reviewer for their valuable time. We explain below how the reviewer's comments have been included in the revised version of the manuscript.

Point-by-point response to Comments

Comments 1: P.2: The main diagnostic criteria are based on DSM-5, and the diagnostic 32 categories remain non-homogeneous, with mechanistic heterogeneity and varying neuropsychological subgroups within each subtype. - While there is no doubt that DSM-V is commonly used in diagnosing ADHD, recently much attention has been focused on Research Domain Criteria (RDoC). I think the authors should at least mention RDOC approach (see Musser & Raiker, 2019 for review)

Response 1: We have added the following sentence about RDoC in lines 41-45

"More recently, a framework of integrated developmental psychopathology (DP) and the National Institute of Mental Health Domain Criteria (RDoC) has attempted to refine theories of ADHD etiology, which may explain heterogeneity at the levels of behavioral and neural circuits\citep{Musser2019}. "

Comments 2: P2: Although individuals with ADHD and typical controls exhibit noticeable differences in performance in numerous tests, there is still considerable controversy surrounding the outcomes of these tests. - It is not clear which controversy the authors are referring to here.

Response 2: Following the reviewer's suggestion, we have explained the controversy in the reviewed version of the manuscript (lines 50-54)

Comments 3: P.2:  By examining high-resolution activity characteristics and standard executive characteristics - I am not sure I understand the distinction between "activity" and "executive" characteristic. In my opinion, substituting "motor" for "activity" could enhance the clarity of the paper.

Response 3: Following the reviewer's suggestion, we have changed the activity to motor.

Comments 4: P.3: Please clarify which groups are you comparing in Table 1 (perhaps in the table title).

Response 4: Following the reviewer's suggestion, we have changed the title from " Demographic characteristics of study participants" to “Comparisons between ADHD-I and ADHD-H/C Groups in Demographic Characteristics”.

Comments 5: P.8: Please clarify which groups are you comparing in Table 3 (perhaps in the table title).

Response 5: Following the reviewer's suggestion, we have changed the title from " Kinect-Based Stroop Color-Word Test by Unadjusted and Adjusted Reaction Time Measures and Group Comparison" to “Comparing Kinect-Based Stroop Test Performance in ADHD-I and ADHD-H/C Groups”.

Comments 6: Figure 7: What do the error bars represent? Please clarify, preferably in the figure caption.

Response 6: Thank you very much for your valuable feedback. Based on our article numbering, we have only five figures. We believe you may be referring to the seventh one when combining figures and tables. We have made modifications to clarify that the error bars actually represent confidence intervals. Additionally, we have revised the captions of the other figures.

Comments 7: Generally speaking, the authors' argument can be improved by incorporating other research that deals with hand motion and ADHD, such as Leontyev et al., 2018.

Response 7: We have incorporated information about this citation in lines 73-78